# Two Facile Aniline-Based Hypercrosslinked Polymer Adsorbents for Highly Efficient Iodine Capture and Removal

**DOI:** 10.3390/ijms24010370

**Published:** 2022-12-26

**Authors:** Biying Liu, Chaochao Mao, Zian Zhou, Qiannan Wang, Xiong Zhou, Zhijie Liao, Ran Deng, Defei Liu, Jingzi Beiyuan, Daofei Lv, Jiesen Li, Liyun Huang, Xin Chen, Wenbing Yuan

**Affiliations:** 1School of Environmental and Chemical Engineering, Foshan University, Foshan 528000, China; 2Foshan Engineering and Technology Research Center for Novel Porous Materials, Foshan 528000, China; 3School of Chemical Engineering and Technology, Hainan University, Haikou 570228, China

**Keywords:** hypercrosslinked polymer, aniline, iodine capture, iodine water, iodine vapor, adsorption mechanism

## Abstract

Effective capture and safe disposal of radioactive iodine (^129^I or ^131^I) during nuclear power generation processes have always been a worldwide environmental concern. Low-cost and high-efficiency iodine removal materials are urgently needed. In this study, we synthesized two aniline-based hypercrosslinked polymers (AHCPs), AHCP-1 and AHCP-2, for iodine capture in both aqueous and gaseous phases. They are obtained by aniline polymerization through Friedel–Crafts alkylation and Scholl coupling reaction, respectively, with high chemical and thermal stability. Notably, AHCP-1 exhibits record-high static iodine adsorption (250 wt%) in aqueous solution. In the iodine vapor adsorption, AHCP-2 presents an excellent total iodine capture (596 wt%), surpassing the most reported amorphous polymer adsorbents. The rich primary amine groups of AHCPs promote the rapid physical capture of iodine from iodine water and iodine vapor. Intrinsic features such as low-cost preparation, good recyclability, as well as excellent performance in iodine capture indicate that the AHCPs can be used as potential candidates for the removal of iodine from radioactive wastewater and gas mixtures.

## 1. Introduction

Nuclear energy is considered a type of renewable and clean energy; however, the challenges and threats of radioactive substances are inevitable in nuclear fuel reprocessing [1,2,3]. Among the radionuclides, radioactive iodine with characteristics of a long radioactive half-life (^129^I: 1.57 × 10^7^ years) can easily access human metabolic systems [4]. The other series of important iodine isotopes is that of short-lived radioactive iodine isotopes, such as ^131^I, ^132^I, and ^133^I, which present the most serious radiological hazards due to external irradiation, inhalation, and ingestion of iodine isotopes. Thus, development of novel materials for treating contaminated water and gas containing radioactive iodine with high efficiency and good durability is of primary importance but is still challenging [5,6,7,8]. 

To remove the iodine radioisotopes, vast adsorbents including activated carbon [9,10], zeolites [11,12], metal–organic frameworks (MOFs) [13,14,15], covalent–organic frameworks (COFs) [16,17,18], and porous organic polymers (POPs) [19,20,21,22,23], have been investigated. However, the low adsorption capacity of silver-based zeolites and the low stability of MOFs under high humidity or in water limit their practical application in iodine capture from gas. Some COFs such as TPB-DMTP COF (620 wt%) [17] and TTA-TTB-COF (500 wt%) [17] demonstrate efficient iodine adsorption in iodine vapor; nevertheless, their large-scale preparation and high cost are big problems. Benefiting from the features of easy scalability, low cost, high chemical/thermal/water stability, and amorphous POPs, hypercrosslinked polymers (HCPs) are especially considered as highly potential adsorbents for the removal of radioactive iodine [24,25]. Zhang et al. [26] prepared highly porous carbon by using triptycene-based HCPs as a precursor and they exhibited excellent iodine adsorption ability of 340 wt% at 75 °C. Li et al. [27] reported a facile one-pot method to construct viologen-based HCPs, with a high uptake ability for iodine vapor up to 525 wt%. Although some progress has been achieved, it is still an urgent task to design and develop simple methods for preparing HCPs to capture iodine efficiently from iodine vapor and iodine water. 

Due to the electron-deficient inherency, the iodine molecule is a superior electron-acceptor and, therefore, the affinity of the adsorbent binding sites to iodine also plays an important role [28]. Volkringer et al. [29] carried out a systematic study of the iodine adsorption performance of functionalized MIL-53 with nine different groups (–H, –CH_3_, –NH_2_, –(OH)_2_, –COOH, –(COOH)_2_, –NO_2_, –Cl, and –Br) and found that the best sorption uptake (60%) was obtained when the ligand was decorated by the electro-donor group –NH_2_. Tesfay Reda et al. [30] prepared a novel composite of lithium aluminate and melamine (LiAlO_2_-Mel) with high iodine adsorption (128.1 wt%), which may be attributed to the N–H⋯I hydrogen bonding and N⋯I halogen bonding interactions. Kobinata et al. [31] measured the dipole moments of the non-bond and charge-transfer complexes between iodine and some aliphatic amines and found that their dipole moments increased significantly with the increment in the amine concentrations. Therefore, the introduction of electron-rich groups, such as amine and imine groups into the polymer structure can greatly enhance the iodine uptake capacity. 

In this context, we aim to develop a new type of adsorbent with rich affinity sites such as –NH_2_ from readily available raw materials. Herein, aniline was chosen to prepare two aniline-based hypercrosslinked polymers (AHCPs), AHCP-1 and AHCP-2, containing rich primary amine groups for effective uptake of iodine. Then, their iodine uptake capacities and adsorption kinetics from the vapor and aqueous solutions were systematically evaluated. The adsorption mechanism was comprehensively described by combining theoretical calculation and these characterizations such as Raman spectra, X-ray photoelectron spectroscopy (XPS), Fourier-transform infrared spectroscopy (FT-IR), and powder X-ray diffraction (PXRD). 

## 2. Results and Discussion

### 2.1. Synthesis and Structural Characterization

The synthesis of AHCP-1 and AHCP-2 were accomplished with a high yield by Friedel–Crafts alkylation and Scholl reactions through aniline as the monomer in the presence of FeCl_3_ and AlCl_3_, respectively (Figure 1) [32,33]. The syntheses were carried out in one step under mild conditions with inexpensive raw materials; therefore, AHCPs were easily synthesized on a large scale. Both AHCPS were insoluble in common organic solvents, water, dilute acids, and alkalis. No signs of decomposition were observed after both compounds were immersed in water, methanol, N, N-dimethylformamide (DMF), DMSO, dilute HCl (3 M), and NaOH (12 M) solutions at 75 °C and 100 °C for a week, indicating that they were chemically stable. 

Moreover, it was found that AHCP-1 and AHCP-2 were thermal stable up to 350 °C and 400 °C, respectively, in air according to their thermo-gravimetric analysis (TGA) spectra (Appendix A). Solid-state ^13^C MAS NMR spectra (Figure 1a) of AHCP-1 and AHCP-2 reveal that the carbon signals (110–150 ppm) were attributed to the benzene ring. The carbon chemical shift near 38.95 ppm was assigned to the carbon in the methylene group linked to the aromatic rings in AHCP-1 [34]. FT-IR spectra (Appendix A) of AHCP-1 and AHCP-2 revealed that the double peak of primary aryl amine characteristic stretching vibrational bands (*v*_N–H_) at 3500–3300 cm^−1^ and bending vibrations (δ_N-H_) near 1620 cm^−1^ were preserved compared with the monomer aniline [35]. For AHCP-1, stretching vibrational bands (*v*_as C–H_) at 2930 cm^−1^ and (*v*_s C–H_) at 2845 cm^−1^ and scissor-like vibrations (δ_C-H_) at 1456 cm^−1^ were preserved compared with dimethoxymethane. The benzene ring possesses stretching vibrations occurring near 1600 cm^−1^, 1560 cm^−1^, 1500 cm^−1^, and 1450 cm^−1^ [22]. Powder XRD measurements confirmed that both AHCP-1 and AHCP-2 were amorphous (Appendix A). The BET specific surface area of AHCP-1 and AHCP-2 as measured by N_2_ sorption were 14.4 m^2^∙g^−1^ and 75.2 m^2^∙g^−1^, respectively (Appendix A and Figure 1b), which were relatively low compared with other porous organic polymers [36]. The value of the BET specific surface area of AHCP-1 was in line with Cooper’s work [33]. The morphologies of AHCP-1 and AHCP-2 networks were examined by SEM (Figure 1c), which indicated aggregated intergrown particles with sizes ranging from hundreds of nanometers to micrometers. Moreover, Figure 1c also showed that the surface of AHCP-1 was smoother than that of AHCP-2, which is consistent with the BET surface area values.

### 2.2. Iodine Adsorption

To evaluate the iodine capture performance of these two aniline-based hypercrosslinked polymers with rich primary amine groups, abundant π-electrons and high water stability, time-dependent UV–vis spectroscopy was carried out to study the iodine uptake of the AHCPs from water. Figure 2a,c showed that upon the addition of AHCP-1, the ultraviolet absorption peak of characteristic iodine decreased rapidly with time. The iodine content decreased greatly from 100% to 8% within 150 min, indicating approximately 92% of the iodine was adsorbed. The color of iodine aqueous solution changed from initial reddish brown to pale yellow after the addition of the AHCP-1 for 30 min, and was nearly colorless after 100 min. As shown in Figure 2b, AHCP-2 also showed a similar color change of iodine adsorbed from water, and its iodine adsorption rate in water was slower than AHCP-1.

The adsorption amount of iodine in AHCPs was monitored by recording sample mass as a function of time after the sample was completely dried (Figure 3a). The AHCPs after iodine adsorption from water were denoted as iodine@AHCPs (water). The iodine uptake capacities of AHCP-1 and AHCP-2 in aqueous solutions at room temperature were estimated to be 250 wt% and 158 wt% respectively, which was superior to other amorphous porous materials reported previously (Appendix A). As shown in Figure 3a, the iodine adsorption capacity of AHCP-1 in iodine aqueous solution was larger than that of AHCP-2, although the latter has a larger specific surface area than that of the former. In addition, AHCP-1 exhibited a faster iodine adsorption rate from water than AHCP-2. These results revealed that the methylene group, as an electron-donating group present in AHCP-1, may be a great aid to increase the interaction between the primary amine group and iodine [29]. This also suggested the dominating factor of the physical adsorption performance of these compounds in aqueous solution was the affinity sites, which has little to do with the specific surface area of the porous absorbent. Considering the presence of a very large amount of water when the adsorbent adsorbs iodine from water, the high affinity sites offered high adsorption selectivity of iodine relative to water molecules [29].

Furthermore, the iodine adsorption performance of AHCP-1 and AHCP-2 in iodine vapor was measured. The iodine adsorption amounts of AHCPs in iodine vapor were monitored by recording sample mass as a function of time at 75 °C or 100 °C. As shown in Figure 3b, the adsorption rates of AHCP-1 and AHCP-2 in iodine vapor were significantly faster at 100 °C than at 75 °C; meanwhile, they reached equilibrium after 20 h and 80 h adsorption, respectively. The saturated adsorption capacities of AHCP-1 and AHCP-2 at 100 °C are 536 and 590 wt%, respectively, and those at 75 °C are 534 and 596 wt%, respectively. This result revealed that the adsorption temperature has a great influence on the adsorption rate, but does not change the adsorption capacity. Compared with AHCP-1, AHCP-2 has higher adsorption capacity (596 wt%), possibly due to its larger BET surface area. Therefore, it can be inferred that the iodine absorption capacity in iodine vapor was not only related to the specific surface area and pore volume of the adsorbent, but also to the number of the affinity sites on the adsorbent [26]. In particular, AHCP-1, AHCP-2 exhibited rapid adsorption, showing an almost linear increment in the initial 15 h and then saturation after 20 h. In contrast, other porous adsorbents, such as TBIM with high adsorption capacity [37], usually required long contact for several days (Appendix A). In addition, the huge difference in the iodine uptake amount in water solution and in the vapor phase for each of the two AHCPs were noticed. This was probably because a large number of water molecules caused a great competition for iodine adsorption, thereby reducing the adsorption capacity of iodine in the iodine aqueous solution. 

Iodine aqueous solutions sorption kinetics were investigated by considering different equations: pseudo-first- and pseudo-second-order kinetic models (Figure 3c and Appendix A). The results showed that the pseudo-second-order kinetic equation can best describe the time dependence for the I_2_ adsorption in aqueous solutions. This model based on the adsorption capacity indicated that the rate controlling step was dependent on a chemisorption phenomenon. The mechanism of chemisorption involved strong interactions between the surface and the adsorbate, confirming the formation of charge transfer complexes between iodine and the AHCPs. It can be seen from Figure 3d that the adsorption of iodine vapor on AHCPs also belonged to the kinetic process of pseudo second order. In order to provide further information about the interactions between iodine and AHCPs, adsorption isotherms were performed at room temperature on AHCPs. Different models have been considered for the interpretation of sorption isotherm curves (Appendix A). Results showed that the Langmuir equation was in good agreement with the sorption curves obtained with AHCPs, indicating a monolayer adsorption process on the surface [38].

Next, a time-dependent UV–vis spectroscopy experiment was performed to study the iodine release of iodine@AHCPs, as shown in Appendix A. It was found that when the solid iodine@AHCPs(water) and iodine@AHCPs(vapor) were placed in methanol at room temperature, the solution color changed quickly from colorless to brownish yellow, then dark red after 40 min of immersion. Furthermore, the result demonstrated that iodine@AHCPs(vapor) released iodine in methanol faster and reached an equilibrium after 80 min, while iodine@AHCPs(water) took about 110 min to reach an equilibrium (Appendix A).

In order to test the recyclability of AHCPs for iodine capture from water and vapor phases, the iodine re-adsorption and desorption experiments for five times using a gravimetric method were carried out. As shown in Figure 4, >80% of adsorption capacity was reproduced in the 5-time adsorption/desorption cycles for AHCP-1 and AHCP-2 in an aqueous solution. The results showed that AHCPs maintained a relatively stable adsorption capacity after five recycles without significant loss. In addition, AHCPs could be recycled at least 5 times while maintaining 70% I_2_ vapor uptake capacity. The partial loss was most likely due to the chemical reaction between iodine and AHCPs at 75 °C and 100 °C [39], namely, the covalent-bonded iodine on the adsorbent skeleton occupied the space where the other guest iodine enters, which leads to the reduction in the adsorption capacity. After these iodination reactions were conducted in the first adsorption, the reabsorption amount of AHCPs tends to stabilize. To confirm this point, the specific surface areas of AHCPs after the first desorption were measured. The results are shown in Appendix A; after the first adsorption–desorption of iodine water and iodine vapor, the BET of AHCP-1 and AHCP-2 became 10.3 m^2^∙g^−1^, 7.1 m^2^∙g^−1^, 63.2 m^2^∙g^−1^, and 46.7 m^2^∙g^−1^, respectively, exhibiting various decline.

### 2.3. Mechanisms 

The outstanding performance of AHCPs on iodine uptake from water solution and from vapor phase encouraged us to investigate the possible mechanism of the iodine adsorption. Various characterization methods, such as Raman spectra, XPS, FT-IR, and PXRD, were performed to detect the interactions between AHCPs and iodine. As shown in Appendix A, a comparison of the Raman spectra among AHCPs and iodine@AHCPs shows that iodine@AHCPs(water) and iodine@AHCPs(vapor) have a similar pattern, suggesting that iodine adsorbed in these AHCPs exists in the same state. All four iodine@AHCP samples exhibit two main peaks around 165 cm^−1^ and 109 cm^−1^; nevertheless, the pristine AHCPs do not. The peaks at ~165 cm^−1^ and ~109 cm^−1^ are assigned to the stretching vibrational modes of polyiodine anions $I_{5}^{-}$ and $I_{3}^{-}$, respectively [40,41]. Comparatively, the peak intensities of iodine@AHCPs(water) at ~109 cm^−1^ are greater than those of iodine@AHCPs(vapor) at ~165 cm^−1^, which may imply that the content of $I_{3}^{-}$ is higher than that of $I_{5}^{-}$ in water. In iodine aqueous solutions, the formation of iodine as $I_{3}^{-}$ and $I_{5}^{-}$ is commonly observed; accordingly, these polyiodine anions can be adsorbed in AHCPs from water. 

Moreover, the XPS spectra were used to analyze the interactions between guest iodine species and AHCPs. As shown in Figure 5a, there were three main groups of peaks around 625 eV, 400 eV, and 285 eV in iodine@AHCP, which were attributed to the binding energies of I3d, N1s, and C1s core levels, respectively. For AHCPs, their XPS spectra were similar to those of iodine@AHCP except for the lack of the binding energy of the I3d core level. Appendix A clearly shows that iodine adsorbed in AHCPs, from water or from vapor, interacted dominantly as I3d_5/2_ and I3d_3/2_ peaks at ~620 eV and ~631 eV. I3d_5/2_ demonstrated the split peaks at 619.0 eV and 620.7 eV in a different ratio. The former can be attributed to $I_{3}^{-}$ and the latter was due to $I_{5}^{-}$. This conformation of the existence of both polyiodine anions in iodine@AHCPs(vapor) and iodine@AHCPs(water) was consistent with Raman spectra. More importantly, a comparison among N1s XPS spectra of AHCPs before and after iodine loading clearly showed the interactions between N atoms and guest iodine species. As shown in Figure 5b, N1s core level peaks of AHCP-1 and AHCP-2 were around 400.3 eV and 399.5 eV, respectively; however, in iodine@AHCP-1 and iodine@AHCP-2, two new peaks around 402.5 eV and 401.4 eV appear. Both new peaks that appeared in iodine-adsorbed AHCPs should belong to the slight shift from 400.3 eV and 399.5 eV arising in pristine AHCP-1 and AHCP-2. This was most likely due to N⋯I and N-H⋯I interactions between nitrogen atoms and iodine species. With regard to C1s in Appendix A, there were no obvious peak changes in AHCPs before and after iodine uptake. All these results clearly showed that the active sites for iodine adsorption in AHCPs were the primary amine groups. Thus, based on previous reports [29,40] and our results here, the presence of these electron-rich groups, such as primary amine and imine groups, can greatly improve the iodine adsorption performance of porous adsorbents.

FT-IR and PXRD were used to study the structural changes of AHCPs before and after iodine adsorption as well. As shown in Appendix A, compared with AHCP-1 and AHCP-2, the N-H scissor-like vibration frequency was blue-shifted by 6 cm^−1^ and 11 cm^−1^ in iodine@AHCPs (water), and blue-shifted by 28 cm^−1^ and 37 cm^−1^ in iodine@AHCPs (vapor) upon complexation with iodine [42]. In addition, it was found that the C-C stretching vibrations (~1600 cm^−1^) of the benzene ring in iodine@AHCPs were red-shifted by 5–22 cm^−1^ compared with AHCPs. All these FT-IR results indicated obvious interactions between AHCPs and iodine [41,43]. In addition, PXRD patterns of iodine@AHCP (Appendix A) did not show any characteristic peaks belonging to molecular iodine under the condition of saturated adsorption, suggesting the transformation of molecular iodine to polyiodine anions as well. These results were consistent with those analyzed through Raman spectra and X-ray photoelectron spectroscopy.

A theoretical calculation can also provide great insights into the adsorption properties in polymers [44,45]. To further reveal the possible main iodine adsorption mechanism, a DFT calculation was performed to investigate the mechanistic reason [44]. Accordingly, the optimized geometry of aniline unit/I_2_ (iodine vapor) and aniline unit/$I_{3}^{-}$ (iodine water) were calculated and selected as shown in Figure 6b,c. For the structural optimization model of aniline unit/I_2_ (iodine vapor), the bond distances of N⋯I and N–H⋯I are, in turn, 2.751 Å and 2.930 Å, and the corresponding binding energy is −0.835 eV and −0.584 eV. The results demonstrated that the binding modes of N⋯I were more stable than that of N–H⋯I. For the structural optimization model of aniline unit/$I_{3}^{-}$ (iodine water), the results proved that the binding modes of N–H⋯I were more stable and have stronger adsorption capacity for $I_{3}^{-}$ (Figure 6c). These results were also consistent with the above characterizations. All the above results certified that the plausible selective adsorption mechanisms were the chelation and electrostatic interactions between iodine and rich primary amine of AHCPs, which are briefly shown in Figure 6a.

## 3. Experimental Section

### 3.1. Preparation of AHCP-1

AHCP-1 was prepared according to the previous reported procedure with a slight modification [32,33]. In a 50 mL round-bottom flask, aniline (2 mmol, 186.3 mg) and dimethoxymethane (8 mmol, 608.7 mg) were mixed together under N_2_ flowing gas. Then, FeCl_3_ (8 mmol, 1297.6 mg) and 1,2-dichloroethane (20 mL) were added. The reaction was then heated to 80 °C for 10 h. After cooling down to room temperature, the precipitate was removed by filtration and extracted extensively by Soxhlet extraction with methanol for 3 h. Finally, the solid was dried under vacuum and at 60 °C overnight. A dark brown powder was obtained (AHCP-1). Yields: 280.7 mg (94%). Elemental analysis of AHCP-1, calcd: C, 80.21%, N, 9.99 %, and H, 9.80%. Found: C, 78.91%, N, 10.6 %, and H, 8.14%.

### 3.2. Preparation of AHCP-2

In a 50 mL round-bottom flask, aniline (2 mmol, 186.3 mg), AlCl_3_ (12 mmol, 1602 mg), and CHCl_3_ (25 mL) were mixed together under N_2_ flowing gas and then refluxed for 10 h. The precipitate was obtained by filtration and subsequently washed with CHCl_3_. The product was further washed with 10% NaOH aqueous solution and distilled water until the pH reached 7. Finally, the product was washed with CHCl_3_ at 70 °C for 3 h. The black solid was dried in a vacuum oven at 60 °C overnight. Yields: 165.6 mg (90%). Elemental analysis of AHCP-2, calcd: C, 77.42%, N, 15.05%, and H, 7.53%. Found: C, 75.78%, N, 13.11%, and H, 6.57%. 

### 3.3. Structural Characterization

The successful preparation of AHCPs adsorbents was confirmed by the following instruments. TENSOR27 Fourier-transform infrared (FT-IR) spectra were collected in transmission on a VARIAN 1000 FT-IR spectrometer using KBr disks. Thermo-gravimetric analysis (TGA) was performed at a heating rate of 10 °C/min under air atmosphere. ^1^H NMR spectrum was recorded on a Bruker Avance III 400 MHz instrument. Solid-state NMR spectrum was recorded on a Bruker Avance III 400 MHz spectrometer by conventional double-resonance 4 mm CPMAS probe at 12 kHz MAS and 298 K with sample packed into a 4 mm MAS zirconia rotor. Scanning electron microscopy (SEM) imaging was performed on a Bruker-BioSpin S-3000N SEM. Low-pressure N_2_ adsorption and desorption isotherms were measured at 77.3 K. The cumulative apparent surface areas for N_2_ were calculated using the Brunauer–Emmett–Teller (BET) model range from 0 to 1.0 bar for all samples. Micropore volumes were calculated using the T-plot method, while the total porous volumes were obtained from the N_2_ isotherm at P/P_0_ = 0.99 and 0.10. Pore size distributions were derived from the N_2_ adsorption isotherms using H-K methods. Powder X-ray diffraction (PXRD) was measured using Bruker-BioSpin D8 Advance-PXRD. X-ray photoelectron emission microscopy (XPS) analysis was performed on a spectrometer (Thermo escalab 250Xi). AHCPs were characterized with Raman spectroscopy having laser excitation energy of 514 nm in inVia Reflex laser Raman spectroscopy.

### 3.4. Iodine Adsorption

Iodine adsorption experiments from water were performed based on the gravimetric method. An amount of 20 mg of AHCPs adsorbents were soaked in 0.1 M (20 mL) of iodine aqueous solution. After iodine adsorption of AHCPs for a given time, the iodine-loaded solid sample was filtered and washed by DI water until the filtrate was colorless, and then dried in vacuum at room temperature for 48 h. The same adsorption experiments were performed in parallel for three times. The iodine uptake of AHCPs samples was calculated according the following equation (m_2_ − m_1_)/m_1_, where m_1_ and m_2_ are the mass of AHCPs samples before and after iodine uptake. 

Iodine vapor adsorption experiments based on gravimetric measurements were performed as the following procedure. An open glass bottle (1.5 mL) containing 25 mg of AHCPs and excess iodine solids were placed in a sealed glass bottle (25 mL), which were then heated at 348.15 K and 373.15 K, respectively. After adsorption of AHCPs in iodine vapor for a while, the bottle was taken out and cooled down to room temperature and then weighted. The same adsorption experiments were performed in parallel three times. The iodine uptake capacity of adsorbents was calculated according to the same equation as the adsorption in iodine aqueous solution. 

### 3.5. Iodine Release

After absorbing iodine of determined mass, place the material in a vial and soak it in methanol at room temperature. Replace the eluent every 2 h until the eluent is colorless, indicating that the desorption is completed.

## 4. Conclusions 

Herein, two aniline-based hypercrosslinked polymers (AHCPs) were successfully developed for industrially cost-effective, efficient, and safe disposal of radioactive iodine from aqueous solution or from their gas vapor. It was notable that AHCPs showed excellent performance in iodine capture including high adsorption capacity, fast rate, and good recycle. Our results suggest that a primary amine group in these aniline-based polymers are crucial for strong iodine adsorption. Considering the low-cost aniline and the facile synthesis, we believe that such aniline-based adsorbents have great potential for iodine removal in nuclear power industry.

## Data Availability

Data is presented in the manuscript.

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
