# Peer review of "Two Facile Aniline-Based Hypercrosslinked Polymer Adsorbents for Highly Efficient Iodine Capture and Removal"

_ijms, 2022, doi:10.3390/ijms24010370_

Round 1

Reviewer 1 Report

In this study, the authors synthesized two aniline-based hypercrosslinked polymers (AHCPs) and utilized those as absorbents for iodine capture in both aqueous and gaseous phases. The authors carried out a variety of experiments for characterizing the structural properties of AHCPs and the absorption performance. The experimental results were clearly presented, providing clear evidence that AHCPs can potentially be used for removal iodine from radioactive waster water and gas. The manuscript was clearly organized and well written. The conclusions were supported by the experiment. Overall, the manuscript is suited for publication in IJMS. 

There is a minor point that might need to be addressed before acceptance for publication. The authors claimed that they designed and synthesized AHCPs. However, there were previous studies synthesizing the AHCP-1 through Friedel-Crafts alkylation reactions. If the authors utilized the same method for synthesis, they should not have claimed that they designed those materials, but instead focus on extending those materials as absorbents for application in iodine capture. 

Author Response

  1. There is a minor point that might need to be addressed before acceptance for publication. The authors claimed that they designed and synthesized AHCPs. However, there were previous studies synthesizing the AHCP-1 through Friedel-Crafts alkylation reactions. If the authors utilized the same method for synthesis, they should not have claimed that they designed those materials, but instead focus on extending those materials as absorbents for application in iodine capture.

Response: Many thanks for the reviewer's comments. Firstly, AHCP-1 was synthesized according to the previous reported procedure with a slight modification, but AHCP-2 was a newly synthesized material by Scholl reactions through aniline as the monomer in the presence of AlCl3. As the reviewer said, our work is focused on extending those AHCPs materials with rich affinity sites as absorbents for application in iodine capture and adsorption mechanism. Therefore, in the sentence description of AHCPs material synthesis, we have appropriately modified “designed and synthesized” to “synthesized” in the revised manuscript.

Reviewer 2 Report

This work studies the polymer adsorbents for highly efficient iodine capture and removal using experimental methods. The manuscript may be accepted for publication after some minor additions/corrections.

1) Introduction: Molecular simulations can also provide great insights into the adsorption properties in polymers (cite e.g., https://doi.org/10.1021/acs.jpcb.9b11840).

2) Abstract: “AHCP-1 exhibits a record-high static iodine adsorption (250 wt%) in aqueous solution”- it will be good to tell here why AHCP-1 has higher adsorption property than AHCP-2 in water, but they are similar for vapor iodine.

3) p.9: “ATCP” -AHCP

Author Response

  1. Introduction: Molecular simulations can also provide great insights into the adsorption properties in polymers (cite e.g., https://doi.org/10.1021/acs.jpcb.9b11840).

Response: We thank the reviewer for the suggestion. We have added the corresponding description and cited the reference mentioned by referee in the revised manuscript (reference [45]).

  1. Abstract: “AHCP-1 exhibits a record-high static iodine adsorption (250 wt%) in aqueous solution”- it will be good to tell here why AHCP-1 has higher adsorption property than AHCP-2 in water, but they are similar for vapor iodine.

Response: Many thanks for the reviewer's comments. The AHCP-1 exhibited faster iodine adsorption rate from water than AHCP-2. This is mainly because the methylene group as an electron-donating group present in AHCP-1, may be a great aid to increase the interaction between the primary amine group and iodine, and thus offers the higher affinity sites. Considering the word limit of the Abstract, we have added the detailed description to in Page 5 (lines 140-145) in the revised manuscript.

  1. p.9: “ATCP” -AHCP.

Response: We thank the reviewer for pointing our mistakes. We have carefully checked the manuscript and corrected it. 
